# Women’s Well-Being and Rural Development in Depopulated Spain

**DOI:** 10.3390/ijerph17061966

**Published:** 2020-03-17

**Authors:** Veronica Cobano-Delgado, Vicente Llorent-Bedmar

**Affiliations:** Departamento de Teoría e Historia de la Educación y Pedagogía Social, Facultad de Ciencias de la Educación, Universidad de Sevilla, C/Pirotecnia s/n, 41005 Sevilla, Spain; cobano@us.es

**Keywords:** well-being, rural development, women

## Abstract

The threat of depopulation in the rural areas making up what has become to be known as “empty Spain” is currently an extremely urgent national issue. Women are a fundamental pillar of rural sustainability, but the lack of decent living conditions has led to their mass exodus to the country’s cities. We analysed the factors undermining their health and well-being, thus leading to their dissatisfaction and their subsequent desire to abandon the countryside for a better life. A mixed methodology was employed, combining qualitative and quantitative data collection techniques. For data collection, an ad hoc questionnaire was developed before being administered to members of the Rural Development Groups of the Celtiberian Highlands, while some of their number were also interviewed. Rural women experience personal dilemmas that prompt them to migrate. These include choosing between living in the place where they were born, close to their families and neighbours, and a decent productive job, the availability of basic services and a broader range of leisure opportunities, among other aspects. It is essential to acknowledge the socio-economic importance of women’s work, to identify invisible burdens and their risks and to adopt measures that facilitate the reconciliation of work and family life.

## 1. Introduction

The Celtiberian Highlands currently have the dubious honour of being the most depopulated region in Spain. There are currently 1355 percent, a municipalities in the Celtiberian Highlands. Despite covering approximately 65,000 km^2^, the area has barely 480,000 inhabitants, accounting for 1 percent of the population of Spain [1].

There are different reasons why women decide to migrate to urban centres, the most frequent being as follows: to seek job opportunities in keeping with their academic qualifications [2,3]; to start a family in a place in which they feel that it is possible to prosper [4]; and to avoid the often dual discrimination to which they are subject in the countryside [5], for being women and for living in such areas [6].

Women living in underprivileged rural areas are more prone to suffer from mental disorders, including depression and anxiety [7] (In this paper, we have used the World Health Organisation’s, hereinafter WHO, definition of mental health as “a state of well-being in which the individual realizes his or her own abilities, can cope with the normal stresses of life, can work productively and fruitfully, and is able to make a contribution to his or her community” [8]. These disorders often derive from feelings of loneliness [9]—by the way, social isolation in rural areas has become an urgent public health issue [10]—difficult access to services, which prevents them from receiving the necessary support [11], and a negative self-perception [12].

This last aspect is primarily related to the social construction of reality that often offers a stereotyped representation of “rural women” [13]. Moreover, it is important to take into account that their occupational integration in these areas is more complex [14], which doubtless affects their well-being and quality of life [15]. In light of this, it should be borne in mind that subjective well-being and satisfaction with life are closely associated with mental health [16]. The WHO recognises the strong link between well-being and health, when asserting that the latter is “a state of complete physical, mental and social well-being and not merely the absence of disease or infirmity” [17].

In short, women face a number of obstacles that make them more vulnerable [18] and which affect their well-being and health as a whole. Since these problems have yet to be described adequately [19], greater attention should be paid to them, while adopting health measures specifically aimed at women and at improving their satisfaction with life. Regarded as key actors in rural conservation and sustainable development [20,21,22,23,24], Spanish rural development policies pay specific attention to women, with a firm commitment to incorporating a gender perspective [25] and to strengthening the analysis of their situation, especially their social and occupational integration [26].

All considered, we believe that it is essential to identify the different situations of rural women [27], insofar as their health is affected by determinants intrinsic to this environment [28] and which contribute to undermine their overall well-being.

## 2. Materials and Methods

The theoretical approach adopted for the explication of health and well-being conditions of rural women has been Maslow’s Hierarchy of Needs. Maslow [29] stated that people are motivated to achieve certain needs and that some needs take precedence over others. Our most basic need is for physical survival (food, drink, clothing, sleep…), and this will be the first thing that motivates our behaviour. Once that level is fulfilled, the next level up is what motivates us, and so on. The following needs in order of hierarchy are: Safety (security, law…); Social (family, friends, belonging…); Esteem (independence, status, prestige); and Self-actualisation (personal growth, authenticity and finding meaning in life…). In conclusion—a desire to become everything one is capable of becoming [30].

In this study, the intention was to seek the views of the members of the Grupos de Desarrollo Rural (hereinafter GDRs) and to analyse the factors favouring the migration of women from the Celtiberian Highlands. The Rural Development Groups are supramunicipal non-profit organisations. Their aim is to implement rural development strategies on the basis of a prior needs analysis [31]. Specifically, the accent was placed on those that, by their reckoning, were having a negative effect on the level of satisfaction and well-being of rural women, and which have been shown to be associated with their mental health.

The idea is to contribute to the sustainability of the most disadvantaged rural area in Spain, taking into account the views of people who have first-hand knowledge and experience of its needs and problems, for the purpose of guiding policies for improving the overall well-being of rural women.

### 2.1. Design

A mixed methodology, combining quantitative and qualitative data collection techniques, was used to inquire into the object of study [32,33]. This method is particularly useful when we want to gain in-depth knowledge of a multidimensional phenomenon that is difficult to analyse using a single research approach [34].

Data collection was performed in three consecutive and complementary stages:(1)A needs analysis conducted on the basis of open interviews.(2)A quantitative analysis performed by administering a questionnaire developed according to the results obtained in Stage 1. The data obtained in this phase had the greatest weight in all the data processing and in general in the whole study.(3)A qualitative analysis based on semi-structured interviews with a view to supplementing the data collected in Stage 2. These data have served to complement the information that was not entirely complete with the questionnaire designed, due to the limitations of the closed-ended items.

### 2.2. Participants

The sample was obtained from the total number of people sitting on the executive boards of the GDRs of the Celtiberian Highlands (chairpersons, vice-chairpersons, secretaries and members) and their technical teams, totalling 280 people, according to the Spanish Network for Rural Development (REDR) [35].

The sample for each one of the data collection stages was as follows:

Stage 1. A total of 14 telephone interviews were conducted with members of the GDRs who had expressed their willingness to participate in the study, this being the point at which theoretical saturation was reached [36], namely, when the information obtained no longer made any new contribution to the needs analysis.

Stage 2. To obtain the sample, the formula for the estimation of proportions for finite populations (*n* = *σ*^2^ · *N* · *p* · *q*/*e*^2^ (*N*
*− 1*) + σ^2^ · *p* · *q*) [37] was employed, with a confidence level of 95 percent and a margin of error of 5 percent, resulting in a sample of 164 subjects. Stratified probabilistic sampling with proportional allocation was used, as shown in Table 1.

Stage 3. A total of 20 interviews were conducted, the point at which theoretical saturation was reached [36].

### 2.3. Instruments

Each one of the instruments was developed ad hoc according to the nature of each data collection stage and the purpose of the collected data.

Stage 1. Open interviews were conducted with the aim of exploring in depth [38] the views of the interviewees on the factors contributing to lower levels of satisfaction and well-being among women living in the Celtiberian Highlands. An inductive approach was employed since it allowed for establishing a series of analytical categories and dimensions, as shown in Table 2, on the basis of the transcriptions of the interviews [39].

These dimensions were employed to develop the questionnaire and the interviews in Stages 2 and 3.

Stage 2. On the basis of the analysis performed on the initial interviews conducted in Stage 1, the questionnaire finally contained five dimensions measured on a four-point Likert scale (1 = strongly disagree and 4 = strongly agree). The questionnaires were completed online using the Google Forms tool, getting a 100 percent response rate.

Cronbach’s alpha was employed to compute the internal consistency and reliability of the questionnaire, obtaining values close to the unit in both the questionnaire as a whole (0.847) and its four dimensions: (1) Work environment (0.865); (2) Services (0.806); (3) Personal sphere (0.808); and (4) Family setting (0.871).

In order to guarantee the construct validity of the questionnaire, the Kaiser–Meyer–Olkin (hereinafter KMO) test and Bartlett’s test of sphericity were employed, obtaining satisfactory values that confirmed the relevance of performing an exploratory factor analysis (EFA). Specifically, this analysis was performed using the principal component method (PCA), generally obtaining saturations higher than 0.30 for each factor, as shown in Table 3.

Stage 3. Lastly, the data obtained from the questionnaires was supplemented by semi-structured interviews. These are especially useful when the interviewer has information on the object of study and the idea is to offer the interviewees the opportunity to discuss the subject more in depth or even to address new issues [40]. These interviews included the following type of questions: How would you describe the current situation in the area with respect to women’s migration? Is there any difference between the work undertaken by men and women? Do you run socio-educational projects for rural women? What socio-educational measures have you proposed to prevent rural women from migrating?

### 2.4. Data Analysis

The qualitative analyses were conducted as follows (Stages 1 and 3) [41]:Data preparation. The interviews, which had been previously recorded with the prior consent of the interviewees, were transcribed.Defining the unit of analysis. The transcriptions were divided into thematic or content units [42,43] pertinent to the study objectives.Developing a category system. A combination of inductive and deductive approaches was employed to analyse the data.Coding with the Atlas-ti computer software (Version 7.2, Scientific Software Development GmbH, Berlín, Germany).

For the quantitative analysis (Stage 2), descriptive (percentages), correlational (Spearman’s ρ for measuring the correlation between ordinal variables) and inferential statistics were used. With respect to the latter, the Kolmogorov–Smirnov (K-S) test was run to confirm that the study sample did not follow the principles of normality. To this end, the Kruskal–Wallis H test and the Mann Whitney U test, both non-parametric methods, were conducted. SPSS ver. 24 (IBM Corp. Released 2016. IBM SPSS Statistics for Windows, Version 24.0. Armonk, NY, USA: IBM Corp.) was used to perform all the statistical analyses.

## 3. Results

The results obtained from the questionnaires and the interviews in Stages 2 and 3 are presented below. The focus is placed here on the four dimensions that, according to the opinions of the experts belonging to the GDRs, affect the well-being of rural women and, consequently, encourage them to migrate to urban areas.

As to the identification data, 60 percent of the respondents were women and 40 percent men. With respect to their professional occupations, 47.9 percent worked in a district (local bodies of a territorial scope created by grouping together several neighbouring municipalities.), 43 percent in the main town of a district (a town that, in the countryside, provides the towns and villages in the surrounding area with services) and 9.1 percent in a city. Moving on to projects, 71.3 percent of the respondents indicated that their GDR had not run any rural development projects aimed exclusively at women, versus 28.7 percent who specified that it had indeed implemented initiatives of this sort.

### 3.1. Work Environment

According to the members of the GDRs, the main reason behind the exodus of rural women was the fact that they had few job opportunities in the Celtiberian Highlands.

To their mind, men were mainly employed in agriculture (97.6 percent) and the building industry (89.1 percent), while women normally had jobs in the tertiary sector (85.5 percent) and public administration (68.5 percent).

A general concern voiced by the members of the GDRs, was the discrimination that women suffered in these areas due to the sexual division of labour. According to the respondents, this meant that women had fewer job opportunities. Specifically, 77 percent of the respondents stated that women tended to start up more small businesses in their villages than men: “Starting up a small business is one of the most frequent professional openings owing to the lack of jobs in other sectors” (Interview 20).

This is striking when taking into account that 75.6 percent of the respondents considered that rural women had higher academic qualifications than men. Moreover, 61.8 percent of them claimed that increasingly more women were pursuing higher education in order to have access to positions of responsibility traditionally occupied by men. However, there were statistically significant differences depending on where the respondents lived. It was observed (Kruskal–Wallis H = 96.607) that those who worked in urban GDRs were, by and large, more in agreement with this tendency (average range = 149.20), in contrast with those who worked in GDRs located in rural districts (average range = 50.93).

In the opinion of 76.4 percent of the respondents, a growing number of women were pursuing higher education to have access to jobs outside their place of residence. Statistically significant differences (*p* = 0.000) were also detected in terms of the respondents’ place of work: in a city, in the main town of a district or in a rural district, as shown in Table 4.

As can be seen, the members of the GDRs who worked in rural districts concurred with this statement more than those working in cities.

Likewise, employing Spearman’s ρ, a statistically significant positive correlation (*p* = 0.000) was observed between the view that a growing number of women were pursuing higher education to have access to jobs outside their place of residence and the impression that they felt inferior, and therefore disregarded, in the workplace (coefficient = 0.478) and the social setting (coefficient = 0.532) in relation to urban women.

With regard to the motivation of rural women to work in the place of residence, there were statistically significant differences (*p* = 0.000 depending on whether or not the GDRs ran rural development projects aimed exclusively at women, as shown in Table 5.

In this sense, in those areas in which the GDRs were running rural development projects specifically aimed at women, most of the respondents concurred that there were increasingly more rural women prepared to work in their place of residence.

It is striking that, despite the job dissatisfaction that the members of GDRs claimed that rural women felt, they did not become proactively involved in the creation of groups with women from other areas with an eye to building bridges (a view expressed by 62.5 percent of the respondents).

Another of the problems detected in the needs analysis was the concern and displeasure that women felt in regard to their national insurance contributions. Nonetheless, 72.7 percent of the respondents held that rural women were becoming increasingly more involved in family concerns in a legal manner (i.e., paying their national insurance contributions).

### 3.2. Services

Rural women’s dissatisfaction and discontent with the available services was, according to the members of the GDRs, the second reason behind their declining level of well-being, a factor that ultimately encouraged them to migrate. The services most in demand on the part of rural women are summarised in the four aspects shown in Figure 1.

The members of the GDRs considered that rural women mainly demanded education and childcare services. All of the interviewees agreed that this issue made it very difficult for mothers living in these areas to reconcile work and family life, thus giving rise to a moral and psychological dilemma in those women forced to choose between having children and a paid job.

About the schools, only 32.1 percent of the respondents believed that the rural schools in their area implemented initiatives aimed at achieving this objective.

Specifically, 73.9 percent of the respondents contended that schools conveyed the idea that students, regardless of their sex, were capable of engaging in any type of employment in their place of residence and, therefore, fostered the professional development of women. Statistically significant differences (*p* = 0.001) were detected in terms of gender (Mann Whitney U = 2418.000). Those women (average range = 74.42) who worked for or sat on the executive boards of the GDRs thought that this idea was conveyed in rural schools to a lesser extent than their male colleagues (average range = 95.86).

According to the respondents, the dearth of new technologies and the inexistence or deficient provision of Internet and mobile telephone services were also behind the dissatisfaction of rural women, especially the young, with the available services. They also stressed that this serious drawback had a particularly negative impact insofar as it prevented access to job opportunities such as freelancing or teleworking, all job modalities in growing demand in the work environment and which would be compatible with living in the countryside.

With respect to health services, the respondents considered that the reduction in the number of doctors per inhabitant and the inexistence of substitutions prevented rural women from accessing these services and put them at a huge disadvantage with respect to city-dwellers.

As to entertainment and leisure services, the members of the GDRs were of the mind that it was an aspect that primarily affected women who wanted to start a family in the countryside. Not only because of the dire implications that this had for the desirable reconciliation of work and family life, but also because the available educational and leisure activities, facilities and initiatives did not cover the requirements that they believed were necessary for their children.

In this regard, there was a general feeling of impotence among the members of the GDRs that were attempting to improve services and to implement projects in their areas but were coming up against financial constraints and external red tape. For this reason, they called for the need for sufficient funding in order to implement them (Interview 8,1).

### 3.3. Personal Sphere

For 70.3 percent of the respondents, women did not see themselves as prime actors in the sustainable development of their villages, especially in regard to helping to promote a much-desired increase in the birth rate. Most of them thought that they were less capable than urban women. The majority of the respondents claimed that rural women felt professionally (77.6 percent) and socially (66.7 percent) inferior to their urban counterparts: “We want the same equal opportunities than in cities. Although not in everything, like a hospital, because it’d be unsustainable, but at least as regards essentials” (Interview 20).

Sixty-six percent of the respondents asserted that rural men had more leisure opportunities than women: “Most women don’t have the wherewithal to amuse themselves. Men go to bars and chat among themselves, watch football, play games … but for women … it’s different” (Interview 19). Statistically significant differences (*p* = 0.000) were observed in terms of the gender of the respondents (Mann Whitney U = 2180.000), with the women agreeing more with this statement than the men (average range = 66.53).

Many rural women had migrated to urban centres in pursuit of greater personal well-being and better career opportunities: “Mothers encourage their daughters to leave their villages for a better life, a job” (Interview 14). They were of the opinion that women, especially the youngest among their number, were given plenty of encouragement to obtain a solid education, “so that they don’t have to depend on anyone, for which reason many of them move to places where there’re adequate education centres” (Interview 20).

### 3.4. Family Setting

Most of the respondents (74.5 percent) considered that those living in the countryside received adequate social support, which often helped them reconcile work and family life: “Certainly, in that sense living in a village has its advantages. Here, everyone knows each other to the point that people can entrust the care of their children to neighbours” (Interview 10). According to 66.7 percent of the respondents, these relationships only made up in part for the problems and difficulties arising from the lack of public services necessary for striking the right balance between work and family life, such as kindergartens, playrooms, schools, etc.

They also claimed that men and women were still pigeonholed in their traditional roles, i.e., the private sphere for women and the public and social spheres for men. Indeed, rural men engaged more often in occupational activities (58.8 percent), while women usually devoted their time to caring for children (68.5 percent) and the aged (70.9 percent). The interviewees concurred that the social pressure to which the sexual division of roles gave rise influenced the way of life of rural women and, for that matter, their diminished worth in the labour market: “Putting an aged relative in a nursing home is usually seen in a more unfavourable light in these areas than in cities” (Interview 14); “In many of these areas, it’s still held that childcare’s a woman’s task” (Interview 13).

So much so that 57.2 percent of the respondents claimed that many women in their area were forced to abandon their jobs in order to devote their time to childcare. They underscored the insurmountable difficulties in reconciling work and family life, mainly due to the dearth of childcare services and subsidies, as one of the main reasons for migrating. As a result, they explained that one member of the couple had to stay at home while the other worked. Owing to their traditional role as housewives and their lower salaries, it was generally women who stayed at home.

### 3.5. Socio-Educational Measures Aimed at Curbing the Exodus of Rural Women

The socio-educational measures that they suggested with a view to improving their living conditions and state of health are analysed and summarised below, as shown in Figure 2.

(1)They contended that schools were crucial for resolving the serious problem of dissatisfaction and the subsequent drop in the female population in the Celtiberian Highlands. These schools are of huge importance for the population, in general, and for women, in particular, basically for:Firstly, they facilitate the reconciliation of work and family life, which is essential for women who, first and foremost, care for children and the aged (Interview 1).Secondly, the lack of schools has a dissuasive effect on men and women who want to start a family in a village, because for them the education of their offspring is a basic service (Interview 19).Lastly, schools as socialising agents play an important role in the non-sexist and non-discriminatory education of women. Education should be a strategic resource for their personal and social emancipation and, by extension, their well-being. According to the interviewees, the first point of intervention should be women’s awareness raising and education (Interview 20). Many women hold that a good education is the only way of feeling valued and appreciated in the society in which they live. Therefore, occupational training programmes that are adapted to the labour demands of the rural area in question and which create more job opportunities should be designed.(2)Improvement of infrastructure. In many cases, the aforementioned deficits in telecommunication services oblige locals to go to a certain area of their villages, usually the highest and most distant point, to access services like 3G (3rd generation). On the one hand, this prevents them from taking distance learning courses or leveraging the opportunities offered by teleworking. Additionally, on the other, due to the lack of broadband services they are unable to download programs, tutorials or training videos, or to handle their own logistics to create and run a company remotely in real time. Technological advances have allowed for integration of rural population into a digital society in which the geographical location of workers is of little importance. With online work, the controversy that the higher the education, the higher the migration level could be surmounted, since “it’s impossible to engage in a job for which you’ve been trained in the rural environment” (Interview 16). With respect to the transport network, the huge distances involved, plus the bad state of the roads, make travelling from one area to another very onerous indeed. By way of example, the lack of adequate road connections turns something as normal as driving to a health centre into an authentic odyssey. In the case of pregnant women or those with children or aged parents (the age groups at greatest risk) in their care, the situation is even more critical: “The mere thought of suffering a serious health problem and not being treated in time is a cause of anxiety for many” (Interview 17).(3)Improving job opportunities and working conditions. The proposals put forward by the respondents and interviewees included tax benefits and positive discrimination measures, plus supporting entrepreneurship and teleworking.

## 4. Discussion

Evidently, rural society has no future without women, for they are indispensable for its continuity and development. The sustainability of rural areas depends, therefore, on encouraging them to remain or settle there [44]. To achieve this, it would be necessary to adopt measures aimed at improving their well-being and health.

Research, like that performed by Gálvez and Matus [45], has placed the accent on the unequal distribution of work between men and women, the latter being assigned the unpaid kind, with little visibility and usually relating to the household and the family, while the former are engaged in paid work.

This social devaluation of their family role means that rural women see themselves as playing a role that is neither very useful nor recognised. Likewise, the inferiority complex that these women tend to have towards their urban counterparts, both professionally and socially speaking, is also a matter of concern. All this seems to bear out the cultural and social perception that people still have of ‘rural or village women’ [13], plus the greater social pressure exerted on them in these areas [46], often lowering their self-esteem and resulting in mental disorders like depression.

Rural women are prevented from having the same job opportunities as men, as has been observed in a number of international reports drafted by the EU [47], in spite of the fact that they tend to possess higher academic qualifications, which just adds insult to injury.

In many cases, moreover, the precarious and temporary nature of women’s employment in these areas means that they are only occasionally employed in productive jobs, which they themselves neither recognise nor value as an additional source of family income, but rather as merely an extra for occasional expenses [45]. As before, their contribution is undervalued and their work is concealed, with their self-esteem suffering as a consequence.

For their part, younger women end up suffering from personal dilemmas due to the contradictions between living in the countryside and having goals and desires similar to those of young urbanites, transmitted and promoted by the current communication society. Their career expectations, which thanks to their academic qualifications, can only be fulfilled in a city, clash with the possibility of remaining in the countryside with their families. This dilemma, together with the frustration that it frequently produces, can lead to serious psychological problems if badly managed [48].

The lack of services and resources characterising many rural areas in Spain [6] prompts many women to migrate to places where they feel that they can get on in the world and start a family more comfortably [4,11].

In short, many women decide to pursue further education with the intention of migrating to a city in search of better job opportunities. This state of affairs could be mitigated by promoting women’s entrepreneurship, which would both better their working conditions and curb their exodus [49].

In relation to the mental health and psychological well-being of rural women, the availability of leisure activities and the wherewithal to participate in them is especially important. It has been shown that feelings of loneliness and isolation, characterising women living in the most disadvantaged rural areas, often lead to mental orders like depression and anxiety [9].

## 5. Limits of the Study and Future Directions

Our research has a number of limitations, which should be addressed in future studies. In this connection, we believe that it would be interesting to seek the views of women actually living in the Celtiberian Highlands, and also of those who have migrated, taking into consideration the adequate statistical parameters for the purpose of supplementing the results obtained in our study. Just as we are aware of the valuable contribution that these women would make with their personal testimonies and life experiences, so too are we equally aware of the problems that this would involve. This limitation results from certain drawbacks of living in the Celtiberian Highlands, including the difficulties in using telecommunication networks to interview these women and in travelling to the region to collect data in situ.

## 6. Conclusions

Considering that ‘improving the well-being of the population’ is one of the objectives of the new health strategy of Europe 2020, we are of the mind that multidisciplinary measures, adapted to each specific context, should be taken with the aim of improving women’s well-being and health in rural areas in evident demographic decline.

In the rural environment, women are faced with personal dilemmas that are difficult to overcome, thus prompting them to migrate. These include choosing between living in their villages, with their families and neighbours, and a decent productive job, the availability of basic services, better leisure opportunities, etc. This is particularly the case when this decision affects their children or future children, who might be deprived of schools, hospitals and better job prospects.

It is essential to acknowledge the socio-economic importance of women’s work, to identify invisible burdens and their risks and to adopt measures that facilitate the reconciliation of work and family life.

There is a need for a greater number of rural development projects aimed specifically at women. As we have confirmed here, these could help them to become more familiar with the options and conditions that the environment offers them and to feel more motivated to live in rural areas in a healthy way.

We also recommend that the sexual division of labour identified here, which is clearly discriminatory towards women, be addressed. Positive discrimination measures that guarantee and promote the health and the occupational, economic, technological, physical, psychological and social well-being of rural women, from a holistic approach, should be adopted.

## Figures and Tables

**Figure 1 ijerph-17-01966-f001:**
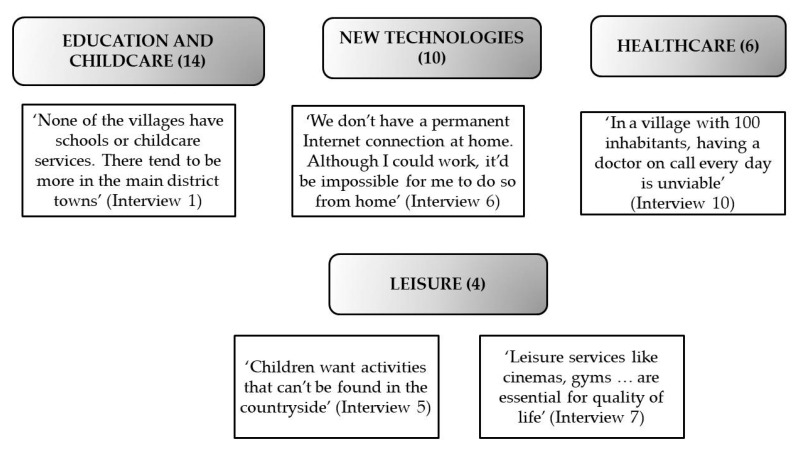
Services and rural exodus. Source: own elaboration. The number indicated in parentheses is the frequency of appearance of the corresponding codes in the transcripts of the interviews.

**Figure 2 ijerph-17-01966-f002:**
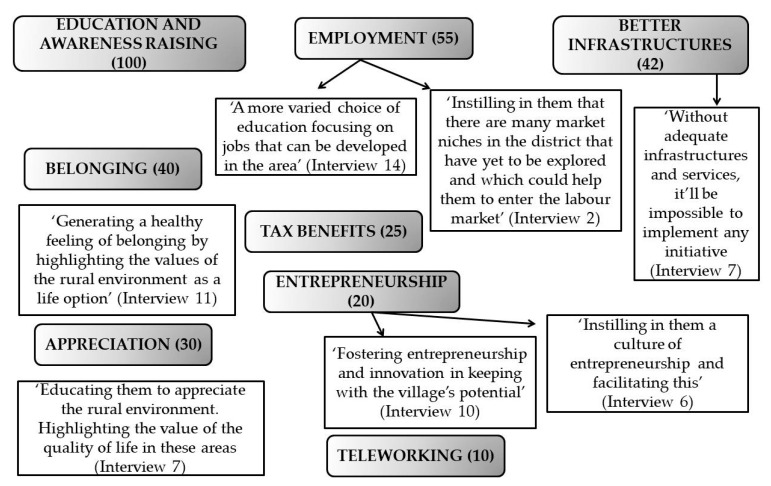
Socio-educational measures for improving the well-being of women in the Celtiberian Highlands, put forward by the members of the Grupos de Desarrollo Rural (GDRs). Source: own elaboration. The number indicated in parentheses is the frequency of appearance of the corresponding codes in the transcripts of the interviews.

**Table 1 ijerph-17-01966-t001:** Population and sample of the quantitative analysis.

Province	No. of Members of the Grupos de Desarrollo Rural (GDRs)	% of the Total	Sample Proportion
Burgos	12	4.29	7
Segovia	23	8.21	14
Soria	18	6.43	11
Guadalajara	56	20	33
Cuenca	56	20	33
Teruel	6	2.14	4
Saragossa	81	28.3	48
Castellon and Valencia	20	7.14	12
La Rioja	8	2.86	5
Total	280	100	165

Source: own elaboration.

**Table 2 ijerph-17-01966-t002:** Analytical dimensions, categories and indicators.

	Categories	Indicators
Identification data	Gender	Male/Female
	Territorial scope	Scope of the GDR
	Projects	Aimed exclusively at women
Work environment	Employment sectors by gender	Agriculture Building Services Administration
	Motivation	Occupational activity in the rural setting Links to other women Entrepreneurship in small businesses Participation in family concerns
	Training	Gender and university education Aim of migration Objective of the traditional occupational activity of men
Services	Education/childcare	Fostering job equality Fostering women’s empowerment
	New technologies	Internet Telephone coverage
	Health	Medical Services Health care
	Leisure	For children Difference men/women
Personal sphere	Self-perception	Contributions to sustainable development Inferiority complex towards urban women in the workplace Inferiority complex towards urban women in the social setting
	Entertainment	Gender and leisure activities
Family setting	Reconciling work and family life	Social support Subsidies/services
	Sexual division of roles	Work Caring for aging parents Caring for young family members
Socio-educational measures for improving the level of satisfaction of women and for favouring sustainable rural development	Training/awareness raising	Needs of the area
		Job opportunities
		Assessment of the area
		Professional training
	Improving infrastructures	New technologies Transport
	Job opportunities	Tax benefits
		Positive discrimination measures
		Entrepreneurship
		Teleworking

Source: own elaboration.

**Table 3 ijerph-17-01966-t003:** The questionnaire’s construct validity.

Dimension	KMO Test	Bartlett’s Test			Factor Loading Ranging (These Values Refer to the Load of Each of the Questionnaire Items on Each Factor, Is Not the Range)	% Variance
		X^2^	gl	Sig.		
1	0.788	986.477	28	0.000	0.792-0.438-0.640-0.813-0.681-0.687-0.847-0.818	52.190
2	0.830	310.629	15	0.000	0.750-0.827-0.723-0.509-0.718-0.775	52.429
3	0.763	620.805	36	0.000	0.620-0.663-0.746-0.654-0.650-0.506-0.581-0.654-0.603	40.151
4	0.846	745.545	66	0.000	0.490-0.618-0.608-0.628-0.597-0.667-0.703-0.736-0.620-0.678-0.701-0.671	41.745

Source: own elaboration.

**Table 4 ijerph-17-01966-t004:** Results of the non-parametric Kruskal–Wallis H test for *k* independent samples.

	Place of Work	Average Range	Kruskal–Wallis H
A growing number of women pursue higher education to have access to jobs outside their place of residence	City	36.50	25.082
	Main town of a district	77.79	
	Rural district	96.51	

Source: own elaboration.

**Table 5 ijerph-17-01966-t005:** Results of the non-parametric Mann Whitney U test for two independent samples.

	Is the GDR Running Any Rural Development Projects Aimed Exclusively at Women?	Average Range	Mann Whitney U
There are a growing number of women inclined to work in the area	Yes	105.53	1667.000
	No	73.25	

Source: own elaboration.

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
