# Peer review of "Women’s Well-Being and Rural Development in Depopulated Spain"

_ijerph, 2020, doi:10.3390/ijerph17061966_

Round 1
Reviewer 1 Report
Dear authors,
From my point of view it is a magnificent article, publishable with minor changes:
Regarding the design of the material and method, some current references could be included to provide scientific support for the mixed method (2. Materials and Methods, 2.1. Design, line 128-129): A mixed methodology, combining quantitative and qualitative data collection techniques, was used to inquire into the object of study [63].
In line 141, put the footnote before the point. It is in Times New Roman. Change.
Materials and Methods", 2.2. "Participants", reads: "Stage 3. A total of 20 interviews were conducted, the point at which theoretical saturation was reached. Can any author be indicated who brings scientific credibility to this question?
In table 2. "Table 2. Analytical dimensions, categories and indicators", some categories that do not provide indicators are provided. They should be specified (Example: New technologies, health, leisure).
In section 2.3. Instruments, it is indicated: The questionnaires were completed online using the Google Forms tool. Should the answer be "100% response rate"?
In the figures in atlas.ti. figure 1, line 305; and figure 2, line 442. Please indicate whether the number in parentheses is the frequency with which these codes appear on the interview transcripts.
Best regards
Author Response
Response to Reviewer 1 Comments
Point 1: Regarding the design of the material and method, some current references could be included to provide scientific support for the mixed method (2. Materials and Methods, 2.1. Design, line 128-129): A mixed methodology, combining quantitative and qualitative data collection techniques, was used to inquire into the object of study [63].
Response 1: We have included the references 33 and 34. Line 214-216:
A mixed methodology, combining quantitative and qualitative data collection techniques, was used to inquire into the object of study [32,33]. This method is particularly useful when we want to gain in-depth knowledge of a multidimensional phenomenon that is difficult to analyse using a single research approach [34]
- Uprichard, E., & Dawney, L. (2019). Data Diffraction: Challenging Data Integration in Mixed Methods Research. Journal of Mixed Methods Research, 13(1), 19-32. DOI: 10.1177/1558689816674650.
- Van Velzen, J.H. (2018). Students’ General Knowledge of the Learning Process: A Mixed Methods Study Illustrating Integrated Data Collection and Data Consolidation. Journal of Mixed Methods Research, 12(2), 182-203. DOI: 10.1177/1558689816651792
Point 2: In line 141, put the footnote before the point. It is in Times New Roman.
Response 2: We have changed the footnote to reference 35. Line 282 and 1078
- Grupos de Acción Local. Available online: http://www.redruralnacional.es/leader/grupos-de-accion-local (accessed on 7 December 2019).
Point 3: Materials and Methods", 2.2. "Participants", reads: "Stage 3. A total of 20 interviews were conducted, the point at which theoretical saturation was reached. Can any author be indicated who brings scientific credibility to this question?
Response 3: We have included the references 36. Line 295:
- Valles, M. S. Entrevistas cualitativas, 2nd ed.; Centro de Investigaciones Sociológicas: Madrid, 2014.
Point 4: In table 2. "Table 2. Analytical dimensions, categories and indicators", some categories that do not provide indicators are provided. They should be specified (Example: New technologies, health, leisure).
Response 4: We have included the indicators of all categories. Tabla 2, Line 304
|
Gender |
- Male / Female |
|
New technologies |
- Internet - Telephone coverage |
|
Improving infrastructures |
- New technologies - Transport |
Point 5: In section 2.3. Instruments, it is indicated: The questionnaires were completed online using the Google Forms tool. Should the answer be "100% response rate"?
Response 5: We have included that clarification in line 319:
The questionnaires were completed online using the Google Forms tool, getting at 100% response rate.
Point 6: In the figures in atlas.ti. figure 1, line 305; and figure 2, line 442. Please indicate whether the number in parentheses is the frequency with which these codes appear on the interview transcripts.
Response 6: We have included the footnote 10, in line 527:
10 The number indicated in parentheses is the frequency of appearance of the corresponding codes in the transcripts of the interviews.
Reviewer 2 Report
Overall, this is a very well performed and systematic study to explore the contributing factors that explains the cause of ‘empty Spain’ in certain rural areas with focused contributions coming from women’s well-being, education, employment, quality of life, availability of basic services and a broader range of leisure opportunities.
This manuscript seems very well-aligned with the special issue of IJERPH, "Potential Risks and Factors of Women's Health Promotion” for the Women's Health section.
I have few comments:
- The introduction section is too long and needs to be shortened to one third of its present word count.
- The results section is too much descriptive and needs to be rewritten in a more concise way.
- The discussion section also appears very much redundant with the results section.
- Citing 79 references in a research article appears significantly high. It would have been better for a review article or a book chapter. The authors are suggested to limit the total number of reference citation under 50.
- ‘Per cent’ used throughout the manuscript needs to be changed to ‘percent’.
Author Response
Response to Reviewer 2 Comments
Point 1: The introduction section is too long and needs to be shortened to one third of its present word count.
Response 1: The introduction had a total of 1551 words, line 26-193. We have reduced it to 494. Thus, we have reduced it to more than a third.
We have also changed the place of a paragraph (lines 195-211), to the device of "2. Materials and methods". We believe that this is more appropriate.
Point 2: The results section is too much descriptive and needs to be rewritten in a more concise way.
Response 2: We have modified all the results. We have reduced information. We have made the writing less descriptive and more concise (line 354-801).
Point 3: The discussion section also appears very much redundant with the results section.
Response 3: We have modified the discussion. We have eliminated information that was redundant and have developed a more concise discussion (line 802-894).
Point 4: Citing 79 references in a research article appears significantly high. It would have been better for a review article or a book chapter. The authors are suggested to limit the total number of reference citation under 50.
Response 4: We have reduced the references to a total of 49 (line 949-1110).
Point 5: ‘Per cent’ used throughout the manuscript needs to be changed to ‘percent’.
Response 5: We've changed the 34 `per cent´ to `percent´ throughout the manuscript.

Reviewer 3 Report
The topic of this article is interesting and relevant for public health, environmental and occupational health, as it deals with a major social phenomenon many European countries has to face: the depopulation of rural areas.
The aim of the study is to investigate the risk factors undermining the well-being and health of women residing in a rural area of Spain leading to their dissatisfaction and their subsequent desire to abandon the countryside for a better life in the city.
The introduction presents the background of the research problem, integrating the major published work in the field. It was adopted a mixed methodology to achieve the aim of the research but is not clear the theoretical approach adopted for the explication of health and well-being conditions of rural women. I suggest to make it clear.
The aim and method sections are clear described, in particular is well explain how the qualitative data of the stage 1 were used to develop the questionnaire.
However, it is not completely clear how quantitative (stage 2) and qualitative data (stage 3) have been integrated. It is not clear whether the qualitative data (stage 3) were used to better explain the data from study 2 which has a greater weight in the data processing or vice versa. Or if the same importance has been given to the two types of data collected. I suggest authors to explain better the process.
The results are presented with enough details to offer to readers the whole picture of the risks that affect the well-being of rural women and, consequently, encourage them to migrate to urban areas. But I suggest to enhance the elements that come from the quantitative data that were strengthened by those qualitative by linking them each other.
I suggest to include the limits of the study and future directions in a separate paragraph.
The References section should follow the criteria of the Journal. I suggest to check that all the citations in the text are reported in the References, to complete with number of pages and DOI where it is possible.
Author Response
Response to Reviewer 3 Comments
Point 1: It was adopted a mixed methodology to achieve the aim of the research but is not clear the theoretical approach adopted for the explication of health and well-being conditions of rural women. I suggest to make it clear.
Response 1: We have defined the theoretical approach adopted for the explication of health and well-being conditions of rural women (line 195-202).
The theoretical approach adopted for the explication of health and well-being conditions of rural women has been Maslow's Hierarchy of Needs. Maslow [29] stated that people are motivated to achieve certain needs and that some needs take precedence over others. Our most basic need is for physical survival (food, drink, clothing, sleep…), and this will be the first thing that motivates our behaviour. Once that level is fulfilled the next level up is what motivates us, and so on. The following needs in order of hierarchy are: Safety (security, law…); Social (family, friends, belonging…); Esteem (independence, status, prestige); and Self-actualization (personal growth, authenticity, and finding meaning in life…). In conclusion: a desire to become everything one is capable of becoming [30].
Point 2: The aim and method sections are clear described, in particular is well explain how the qualitative data of the stage 1 were used to develop the questionnaire.
However, it is not completely clear how quantitative (stage 2) and qualitative data (stage 3) have been integrated. It is not clear whether the qualitative data (stage 3) were used to better explain the data from study 2 which has a greater weight in the data processing or vice versa. Or if the same importance has been given to the two types of data collected. I suggest authors to explain better the process.
Response 2: We have explained the importance given to quantitative data and qualitative data (line 271-277), and how to integrate both.
- A quantitative analysis performed by administering a questionnaire developed according to the results obtained in Stage 1. The data obtained in this phase had the greatest weight in all the data processing and in general in the whole study.
- A qualitative analysis based on semi-structured interviews with a view to supplementing the data collected in Stage 2. These data have served to complement the information that was not entirely complete with the questionnaire designed, due to the limitations of the closed-ended items.
Point 3: But I suggest to enhance the elements that come from the quantitative data that were strengthened by those qualitative by linking them each other.
Response 3: We have modified all the results. We have reduced information. We have made the writing less descriptive and more concise (line 354-801).
Point 4: I suggest to include the limits of the study and future directions in a separate paragraph.
Response 4: We have included the limits of the study and future directions in a separate paragraph: `5. Limits of the study and future directions´(line 895-904).
Point 5: The References section should follow the criteria of the Journal. I suggest to check that all the citations in the text are reported in the References, to complete with number of pages and DOI where it is possible.
Response 5: We have reduced the references to a total of 49 due to an indication from another reviewer.
In addition, we have checked that all the citations in the text are reported in the References, and we have completed reviewed and changed with number of pages and DOI (line 949-1110),
Round 2
Reviewer 2 Report
I am satisfied with the answers provided by the authors in response to my queries and concerns. The revised version of the manuscript has been improved significantly.